# Performance Analysis of Geiger–Müller and Cadmium Zinc Telluride Sensors Envisaging Airborne Radiological Monitoring in NORM Sites

**DOI:** 10.3390/s20051538

**Published:** 2020-03-10

**Authors:** Jorge Borbinha, Yuriy Romanets, Pedro Teles, José Corisco, Pedro Vaz, Diogo Carvalho, Yoeri Brouwer, Raul Luís, Luís Pinto, Alberto Vale, Rodrigo Ventura, Bruno Areias, Andre B. Reis, Bruno Gonçalves

**Affiliations:** 1Centro de Ciências e Tecnologias Nucleares, Instituto Superior Técnico, Universidade de Lisboa, Estrada Nacional 10, ao km 139,7. 2695-066 Bobadela, Portugal; yuriy@ctn.tecnico.ulisboa.pt (Y.R.); ppteles@ctn.tecnico.ulisboa.pt (P.T.); corisco@ctn.tecnico.ulisboa.pt (J.C.); pedrovaz@ctn.tecnico.ulisboa.pt (P.V.); 2Instituto de Plasmas e Fusão Nuclear, Instituto Superior Técnico, Universidade de Lisboa, Av. Rovisco Pais 1, 1049-001 Lisboa, Portugal; diogo.d.carvalho@tecnico.ulisboa.pt (D.C.); ybrouwer@ipfn.tecnico.ulisboa.pt (Y.B.); rluis@ipfn.tecnico.ulisboa.pt (R.L.); lpinto@ipfn.tecnico.ulisboa.pt (L.P.); avale@ipfn.tecnico.ulisboa.pt (A.V.); bruno@ipfn.tecnico.ulisboa.pt (B.G.); 3Institute for Systems and Robotics, Instituto Superior Técnico, Universidade de Lisboa, Av. Rovisco Pais 1, 1049-001 Lisboa, Portugal; rodrigo.ventura@isr.tecnico.ulisboa.pt; 4Instituto de Telecomunicações, Universidade de Aveiro, 3810-193 Aveiro, Portugal; brunoareias@ua.pt (B.A.); abreis@ua.pt (A.B.R.)

**Keywords:** unmanned aerial vehicles, radiological monitoring, gamma spectrometry, uranium contamination, NORM

## Abstract

Radiological monitoring is fundamental for compliance with radiological protection policies in the aftermath of radiological events, such as nuclear accidents, terrorism, and out-of-commission uranium mines. An effective strategy for radiation monitoring is to use radiation detectors coupled with Unmanned Aerial Vehicles (UAVs), enabling for quicker surveillance of large areas without involving the need of human presence in the target area. The main aim of this study was to formulate the parameters for a UAV flight strategy in preparation for future field measurements using Geiger–Muller Counters (GMC) and Cadmium Zinc Telluride (CZT) spectrometers. As a proof of concept, the prepared flight strategy will be used to survey out-of-commission uranium mines in northern Portugal. Procedures to assure the calibration of the CZT and verification of the GMCs were conducted, as well as a sensitivity analysis of the sensors considering different acquisition times, distance to source, and detector response time. This article reports specific parameters, such as UAV distance to ground, time of exposition, speed, and the methodology to perform the identification and calculate the activity of possible radioactive sources. An effective flight strategy is also presented, aiming to use radiation detectors coupled with UAVs to undertake extensive monitoring of areas with enhanced levels of environmental radiation, which is of prime importance due to the lasting hazardous effects of enhanced environmental radiation in the nearby ecosystem and population.

## 1. Introduction

### 1.1. Contextualization

The purpose of radiological monitoring is to verify compliance with radiological protection policies according to international recommendations. The most serious scenarios occur in the aftermath of severe radiological events, such as radiological dispersion device outbreaks (“dirty bombs”) and nuclear accidents—the most prominent examples being the accidents at the Chernobyl Nuclear Power Plant in 1986 and at the Fukushima Daiichi Nuclear Power Plant (FDNPP) in 2011 [1,2]. Nevertheless, the surveillance of continental areas where uranium mining has taken place is also a very important aspect of radiation monitoring, since the incident radiation typically encountered during routine radiation surveys is a consequence of naturally occurring radioactive materials (NORM). Uranium mines entail the production of large amounts of solid wastes with enhanced concentrations of natural radionuclides from the U and Th decay series, which are classified as NORM [2,3,4].

Between 1908 and 2001, 60 deposits of radioactive ore were extracted for radium and uranium production in Portugal. These mines, which are predominantly in northern interior Portugal, exhibit high levels of background radioactivity. All mining and milling activities ceased in 2001, when the Portuguese government recommended the “Study of Effects of Uranium Mining Residues on the Public Health” project (MinUrar Project). Among other results, this project showed high concentrations of the uranium family of radionuclides at the mining and milling tailings at different locations. At some locations, ambient radiation doses reached values of 41 mSv/year on uranium tailings. Unequivocally, the uranium milling tailings are a radiation source originating dose rates which substantially exceed not only the regional radiation background but also the recommended (Directive 96/29/EURATOM) additional dose limit of 1 mSv/year to members of the public [5].

The dangers of radiation contamination in the environment, such as uranium mining tailings, have become a prominent issue because they entail hazardous risks to humans, as well as to the fauna and flora of the area. Therefore, it is of cornerstone importance to protect the population from ionizing radiation. Radiation mapping aims to locate, identify, and quantify the intensity of a source in a radiation exposure scenario, by using effective measurement equipment, such as radiation detectors [3,4,6,7].

### 1.2. State of the Art

Radiation mapping can be either ground-based or airborne. Ground-based mapping can be performed using static or mobile radiation detectors, such as handheld or vehicle-mounted detectors [3,6,8]. Historically, airborne radiation mapping has been performed using large crewed aircraft coupled with large volume detectors [3,6]. Although this method presents some advantages, such as high operational speed and possibility to map otherwise inaccessible areas, there are also drawbacks, such as low spatial resolution (due to the high altitude at which this mapping method operates) and high initial cost [3,4,6].

The development of Unmanned Aerial Vehicles (UAVs) has outdated much of the need for piloted radiation monitoring. Although radiation monitoring using UAVs is limited by the flight time and payload, it presents many advantages, such as improved spatial resolution (mapping at lower altitude than larger aircraft), faster data collection and greatly reduced risk of exposure to professionals. Furthermore, it is considerably less expensive than larger aircraft and not limited by terrain irregularities [3,4,6]. Despite having been applied to radiation monitoring earlier [9,10], UAVs have suffered a major scientific and technological boost since the accident at the FDNPP in 2011 [4,6,11,12,13,14,15,16,17,18,19,20]. UAVs may have fixed or rotary wing. Rotary-wing UAVs are usually constructed with multirotor pairs and thus are also known as multirotors or “drones”. Moreover, rotary-wing UAVs have the advantage of being able to do “loitering” at a short distance over specific targets [20].

UAVs have been coupled with radiation detectors in many previous studies for the purposes of radiological inspection or monitoring in several worldwide locations [3,6]. Geiger–Muller Counters (GMCs) were used in early studies of UAV radiation mapping [9]. GMCs are gas-filled tubes, on which the incident radiation ionizes the gas molecules, forming negative and positive ions that are attracted towards the detector’s cathode and anode, respectively. This process triggers electronic pulses that reflect the count of detected particles. GMCs are easy to use, lightweight, and have large sensitive volumes. However, their low count efficiency and inability to acquire radiation spectra has caused GMCs to be disregarded in UAV radiation monitoring since the work of Kurvinen et al. [6,9].

UAVs have also been coupled with scintillator detectors, which are spectrometers (able to differentiate the incident radiation as a function of energy). However, due to their low spectral resolution and poor peak shaping, scintillators are not recommended for detecting low-level radiation [3,6,9,18].

Semiconductor detectors are an alternative to scintillators. Semiconductors work by creating electron–hole pairs when photons interact with the sensitive volume. Electron–hole pairs separate when a voltage difference is applied to the cavity and the signal is then measured using charge sensitive detectors [21,22,23]. Semiconductors are known for their excellent spectral resolution and counting efficiency, allowing for accurate and precise radiation measurements. Although other good semiconductor materials are available (e.g., silicon and high-purity germanium), cadmium zinc telluride detectors (CZT) present many advantages for gamma-ray detection, such as being compact, portable, and able to be used at room temperature and above without a cooling system. CZTs have, however, one major problem: the issues with the crystal growth process, which increase production costs and limit the detector volume to small sizes, in the order of 1 mm${}^{3}$ [6,21,22,23]. Several studies have emerged in the past few years coupling CZT detectors with UAVs to perform radiological inspection/monitoring in areas near Fukushima [13,14,15,17] or even in out-of-commission uranium mines [4]. Therefore, considering the extensive testing and analysis effort that has been made, the application of UAVs coupled with radiation detectors (such as GMCs and spectrometers) is a promising approach to map the radioactivity levels and radionuclide identification in areas of interest. Moreover, advances in multirotors and in the ability to “loiter” are particularly useful for radiological inspection or monitoring because they allow for GMC transport near the target and for landing and take-off in specific locations with the CZT [20].

The effects caused by ionizing radiation in electronic devices, such as the limitation of the performance and survival of devices, pose a serious concern [24,25]. Namely, this effect could be observed in the UAVs used to survey the FDNPP after the accident in 2011, in which the UAVs would malfunction much more quickly than expected [26]. The external radiation dose rates on uranium tailings are typically above background radiation levels. Nevertheless, these levels are minimal when compared to the dose rates needed to produce observable effects on electronic components even if the UAV spends many hours flying in proximity to the tailings. Therefore, also taking into account the low concentrations and activities of radionuclides expected to be found in out-of-commission uranium mines, the authors consider that the radiation effects on the UAVs will be minimal and will not compromise the mission.

### 1.3. Aims and Scientific Contributions

The main goal of this work is to set up a flight strategy in preparation for future field measurements using GMCs and a CZT spectrometer coupled with a fleet of drones in continental areas where radioactivity levels are expected to overcome the natural background, due to NORM industries or radiological events. As a proof of concept, the prepared flight strategy will be used to survey out-of-commission uranium mines in northern Portugal.

A calibration of the CZT and a verification of the GMCs were conducted in order to assure the correct measurement of equivalent dose rate values, as well as an accurate identification and activity estimation of the source. In addition, a detector sensitivity analysis was performed for the GMCs and the CZT. Therefore, this study also aimed to specify the most appropriate physical parameters of the UAV flight (distance to the ground, time of exposition, UAV speed, hover time, etc.) for radiological data gathering, considering the characteristics of the sensors and location (out-of-commission uranium mines) of the field measurements prepared in this study. Furthermore, this work reports the protocol to, in a field scenario, use the raw data provided by the sensors to determine the location of the source, as well as the identity and activity of the radionuclides present in the soil.

The fundamental calibration procedures of radiation detectors are well-known. CZTs have been calibrated in other works [27,28,29,30,31] and even coupled with UAVs for radiological inspection/monitoring purposes [4,13,14,15,17].

There are essentially two scientific contributions of this paper. Firstly, to the best of the authors knowledge, there are no other published works reporting verification/calibration and thorough performance analysis (reporting physical parameters such as distance to ground, hover time, acquisition time, and UAV speed) for the purpose of coupling GMC and CZT sensors with a fleet of drones for radiation monitoring in regions with high NORM concentrations. Secondly, another novelty of this work lies in the proposal and analysis of a flight strategy combining the use of GMC and CZT sensors coupled with cooperative navigation of a fleet of drones to perform radiological monitoring and mapping in areas with high concentrations of NORM (i.e., out-of-commission uranium mines).

This paper is organized as follows. Section 2 describes the framework in which this work was developed and the proposed approach for a UAV flight strategy for radiological inspection and monitoring. Section 3 outlines the materials used and procedures applied to perform the calibration of the CZT and the verification of the GMCs, as well as the detector sensitivity analysis. Section 4 presents the results of the calibration and verification procedures and of the sensitivity analysis. Furthermore, a discussion of the results is presented focusing on the outcomes of this work and on the consequent recommendations and flight strategy for radiological monitoring using UAVs. Section 5 presents the main conclusions of the study.

## 2. Framework and Flight Strategy

This work was performed in the scope of the project FRIENDS (Fleet of Drones for Radiological Inspection, Communication and Rescue). The objectives of the project FRIENDS are to design, develop and validate a fleet of UAVs equipped with navigation and radiological sensors for inspection and monitoring of hazard scenarios, such as nuclear terrorism, accidents, or areas with high concentration of NORM. This project addresses this challenging problem by proposing a fleet of UAVs to perform this characterization. In particular, the project focuses on three challenges depicted in Figure 1: (1) real-time mapping of the scenario and contamination level of an infected area; (2) autonomous and cooperative navigation of a fleet of UAVs; and (3) wireless data communication between the UAVs and the ground station [32].

The detectors available for the FRIENDS project were the μSPEC 500 CZT spectrometer, henceforth denoted CZT, and two GMCs, a Sparkfun 11345 and a Mazur PRM-9000, henceforth denoted Sparkfun and Mazur, respectively. The CZT was chosen because it is compact, able to work at room temperature, and has an excellent counting efficiency. The Mazur is appropriate for the measurement and monitoring of low energy radionuclides, particularly NORM. The Sparkfun was tested as a lower cost alternative to the Mazur. The radiation sensors used in this work are shown in Figure 2.

In the scope of the project FRIENDS, the proposed flight strategy for the fleet of drones can be formulated in five stages, in each an agent of the fleet acquires different data that will contribute to map the scenario:**Stage 0**: At the base station, the area of interest to be monitored is defined and the drones are dispatched.**Stage 1**: The LIDAR Drone scans the area of interest and performs 3D reconstruction to provide spatial information and help the other agents to navigate.**Stage 2**: The GMC-drone performs a complete coverage of the area of interest to acquire the equivalent dose rate values along the flight path. At this stage, the construction of the radiological map of the area currently being monitored begins and it is possible to identify points with enhanced equivalent dose rate levels, when compared to the background radiation (i.e., hotspots). In this phase, it is only known whether the equivalent dose rate at that location is above background level or not.**Stage 3**: The GMC Drone hops between the hotspots identified in Stage 2, landing in each one during a preset amount of time necessary to perform precise verification of the presence of radioactive sources and acquire the real equivalent dose rate value.**Stage 4**: The CZT Drone hops between the hotspots, landing in each one and acquiring the spectrum of the present gamma emitters. The analysis of the acquired spectrum allows for estimation of the identity and activity of the sources located in the hotspot. This agent lands for a preset amount of time and the CZT will be at a distance from the ground that optimizes the acquisition of a low uncertainty spectrum.

## 3. Materials and Methods

The calibration measurements were performed using a USB-powered CZT spectrometer (detector characteristics can be consulted in Table 1. Data acquisition and configuration was performed using a MATLAB script. This script was connected to the CZT and returned data matrices containing the accumulated number of counts for each channel, in preset time intervals and during a predefined total acquisition time. The spectra information was then later imported and analyzed using the open source software InterSpec v.1.0.1a [33].

In addition, two GMCs were also coupled to the UAVs: the Sparkfun and the Mazur (Figure 2). Table 2 presents some characteristics for both. A verification procedure was undertaken using both sensors, in order to assure the accuracy of the quantities that the sensors provide. Data acquisition was performed using the MARIA [34] software application. When the GMCs are connected to a smartphone, MARIA presents the data in CPM (counts per minute) in time intervals that can be set by the user and stores the data in a file. This also contains information about time and location of each measurement.

### 3.1. Energy and Efficiency Calibration

The counting system needs to be calibrated in three different ways in order to analyze gamma radiation emission spectra: (1) energy calibration; (2) calibration of the shape of peaks, meaning full width at half-maximum (FWHM); and (3) calibration in efficiency.

To obtain a sufficient amount of peaks within the energy range, two certified calibration sources were used: (1) a Eu-152 source (calibrated on 30 March 2006); and (2) a mixed gamma aqueous solution (calibrated on 1 April 2013). The acquisition conditions used for calibration were the same as for weekly quality control. The CZT was placed with a Source Detector Distance (SDD) of 10 cm, on top of a holder with enough height to line the center of the detector with the center of both calibration sources, which were attached with duct tape. The acquisition times were 86,400 s and 284,500 s for the foreground and the background spectra, respectively. The activity of the sources (number of particles emitted by the sources per unit of time) at the time of measurement was calculated using the radioactive decay law, considering the activity (A) and half-life ($t_{\frac{1}{2}}$) information present in the calibration certificates [37].

Table 3 shows the energy peaks emitted by the calibration sources. Figure 3 represents the acquired calibration spectrum. Not all peaks were considered for calibration, for various reasons: (1) the peak is unidentifiable by the software; (2) the peak is the summation of two or more peaks; and (3) the peak has too low efficiency. Since many of the nuclides in the mixed gamma source had short half-life, it was not possible to acquire enough counts in the spectra to correctly estimate their activity. InterSpec was used to calculate the peak mean energies, full width at half maximum (FWHM) and areas, as well as to identify the peaks. In total, 12 peaks (marked with an “*” in Table 3 and labeled in Figure 3) were considered, with energies between 59 keV and 1408 keV.

### 3.2. Quality Assurance

To check the accuracy of the obtained calibration curves, three radiation sources independent from the sources used in the calibration were used: Pb-210, Cs-137, and Co-60. The acquisition setup was the same as for calibration. For the validation, the acquisition time was 5020 and 15,000 s for the foreground and the background spectra, respectively. Mean energies were calculated for each peak and compared with the certificate peak energies.

For each peak, the activity was estimated using the peak area (measured using InterSpec). The efficiency value for each peak was retrieved from the efficiency calibration curves, according to the peak mean energy. Then, the obtained activity value was divided by the yield of the peak in the certificate to obtain the nuclide activity in Bq, which was compared to the activity values present in the respective calibration certificates.

### 3.3. CZT Sensitivity Analysis

A sensitivity analysis of the CZT detector was also performed, by studying the detector response with varying SDD and acquisition time. This analysis aimed to provide specific recommendations related to UAV measurements, such as time required for reliable spectrum acquisition (low uncertainty), distance from UAV to ground, and hover time.

Using both calibration sources and the same acquisition geometry, spectra were acquired with the CZT detector with SDD of 10, 100, and 200 cm. The acquisition time was 1800 and 5400 s for the foreground and the background spectra, respectively. In addition, using both calibration sources and the same acquisition geometry, spectra were acquired with the CZT detector at SDD of 10 cm, with acquisition times between 1 and 30 min.

### 3.4. GMC Verification

The verification of the GMC sensors was performed at the *Laboratório de Metrologia da Radiação Ionizante* (LMRI), *Campus Tecnológico e Nuclear (Pólo de Loures), Instituto Superior Técnico*. The LMRI is a certified primary standard laboratory of the Portuguese National System of Metrology that performs activities such as quality control of devices that measure ionizing radiation.

Each of the GMCs was placed at three different distances (100, 150, and 200 cm) from three different Cs-137 sources with different activities (74, 740, and 7400 MBq) and tabulated equivalent dose rate values. The UAVs are expected to fly at these distances from the ground during a field measurement, mainly because of the vegetation height. The geometric center of the sources was aligned with the geometrical center of the sensitive volume of each GMC using vertical and horizontal lasers (Figure 4). Nine data acquisitions were performed, one for each source-distance configuration. A set of twenty values (in CPM) was retrieved for each source-distance configuration and background, with intervals between readings of 10 s for the Mazur and 5 s for the Sparkfun. For each source-distance configuration, the average of the acquired set of values was calculated and the average background value was subtracted from it. Moreover, the standard deviation, coefficient of variation (ratio of the standard deviation to the mean), and conversion factor (ratio of the source tabulated equivalent dose rate value to the average for each source-distance configuration) were also calculated.

### 3.5. GMC Sensitivity Analysis

A sensitivity analysis of the GMCs was also performed, by studying sensor response with varying SDD and acquisition time. This analysis aimed to provide specific recommendations regarding the flight strategy for the UAVs carrying GMCs, namely distance to ground, time of exposition and UAV speed. Two certified Cs-137 sources were used for this procedure, with a combined activity of approximately 0.143 MBq at the time of measurement. The activity of the sources at the time of measurement was calculated as described in Section 3.1 using the radioactive decay law. Firstly, each GMC was placed at three different distances of both Cs-137 sources (20, 50, and 80 cm) and the reading of each GMC was acquired (in CPM) at acquisition times of 1, 2, 5, 10, 30, and 60 min. In total, 18 data acquisitions were performed, one for each source-distance configuration. The sensitivity of each GMC was studied at lower distances, due to the lower activity of the sources that were available for the sensitivity analysis. Furthermore, the lower activity sources are a more accurate approximation to the conditions and sources that are expected in a field scenario.

In addition, the response time was studied for each GMC. Both radiation sources were placed at 3 cm from the GMCs with the geometric center of the sources aligned with the geometrical center of the sensitive volume of the GMC. A chronometer was used to measure the time between the placement of the sources in front of the GMCs and the count rate raising higher than a set threshold. This methodology was repeated 20 times for each GMC. The average, standard deviation, and coefficient of variation were calculated to determine the response time of each GMC. Moreover, since the shortest time interval in which the Mazur provides measurements is 5 s, the time interval was also set to 5 s for the Sparkfun in order to have a coherent methodology for both GMCs. Nevertheless, this methodology was also applied for the Sparkfun considering a shorter time interval of 1 s.

Given the high coefficient of variance exhibited in the Sparkfun measurements, a higher count rate is necessary in order to obtain a clear distinction between the background count rate and the count rate with the presence of a source in just a few seconds. Therefore, the 3 cm SDD was chosen not only because of the lower combined activity of the sources (compared with the sources used in the GMC verification), but also due to the Sparkfun showing an average count rate that is clearly distinguishable from the values usually measured from background. The count rate threshold considered was calculated as the average between the count rate measured with the sources at 3 cm and at background conditions (which were calculated as the average of 20 measurements taken in 5 s intervals).

## 4. Results and Discussion

### 4.1. Energy and Efficiency Calibration

The energy calibration function converts the channel number *c* to energy. In semiconductor spectroscopy, energy calibration is typically represented by a linear regression, defining the relationship between channel number and measured energy of the peak mean energy. The full FWHM is another metric used in spectroscopy. Shaping, or FWHM, calibration allows for the correct adjustment of experimental peaks, which is mandatory for the integration of total counts inside the peak area. A linear regression of the FWHM values was calculated. Efficiency (ϵ) was calculated as the ratio between the number of counts measured by the detector and the total number of particles emitted by the sources (i.e., activity of the sources at the time of measurement), in a set amount of time. The number of counts measured by the detector was calculated as the peak area, as presented by *InterSpec*. The efficiency uncertainty was retrieved directly from InterSpec. Table 4 presents the peaks considered for calibration, as well as their characteristics taken from the CZT spectrum acquired with the calibration sources (Figure 3).

To obtain the energy calibration curves, Microsoft Excel${}^{TM}$ was used to obtain a least-squares linear fit of the data. For the FWHM curve a square root dependency was considered. The energy calibration measurements and linear fit are depicted in Figure 5 and can also be described by Equation (Equation 1). The FWHM calibration curve and linear regression, described by Equation (Equation 2), can also be observed in Figure 5.
(1)E(keV)=1.759·c−0.0779(R=1)
where *c* is the channel number, and *E* the energy in KeV.
(2)$$FWHM\left(keV\right)=0.3833\cdot \sqrt{E}+1.2253\;\left(R=0.8886\right)$$

The efficiency value for each considered peak, as well as its uncertainty, can be consulted in Table 4.

Two polynomial regression curves of the logarithms of energy and efficiency were calculated: one for lower energies and one for higher energies. To obtain the efficiency calibration curves, Microsoft Excel ${}^{TM}$ software was used to obtain third- and second-order polynomial fits of the data. A logarithmic dependency was considered for both the dependent variable (energy) and the independent variable (ϵ). Figure 6 shows peak efficiency for each considered peak.

Up to 344.3 keV, the calibration curve is given by:(3)$$ln\left(\epsilon \right)=543.53+243.85\cdot \ln\left(E\right)+35.297\cdot \ln{\left(E\right)}^{2}-1.7754\cdot \ln{\left(E\right)}^{3}\;\left(R=0.997\right)$$

Above 344.3 keV, the calibration curve is:(4)$$ln\left(\epsilon \right)=-30.595+10.249\cdot \ln\left(E\right)-1.1377\cdot \ln{\left(E\right)}^{2}\;\left(R=0.999\right)$$
where ϵ is photopeak efficiency and E is the peak energy.

Several fits were obtained for the data at different energy ranges. Equations (Equation 3) and (Equation 4) correspond to the fits that minimized the relative differences between the measured efficiencies of the peaks and the efficiency values calculated with the fit curves. The relative differences between the measured efficiencies of the peaks and the efficiency values calculated with the fit curves were, on average, 0.3%, with maximum and minimum values of 0.6% and 0.04%, respectively.

The manufacturer declares the resolution of the CZT detector to be better than <2.5% FWHM @ 662 keV. The measurement performed in this work for the same energy and described geometry shows 8.8 keV (1.33%) FWHM, which is better than expected. Moreover, the efficiency calibration curves observed in Figure 6 were compared to other published efficiency calibration curves of CZT detectors [27,28,29]. The efficiency curve in this work is very similar in shape to the other published efficiency curves, whereas the values of the efficiencies in other works vary one or two orders of magnitude, which may be explained by the use of different acquisition geometries and different models of CZTs.

The calibration curves reported in this work were obtained considering peaks between 59 and 1408 keV. When performing measurements in areas with high uranium concentrations, such as out of commission uranium mines, it is expected to find radioisotopes created from the decay of U-235, U-238, and Th-232. The gamma emissions of the decay of these radionuclides are mainly within the range of energies used in this work. No list of the emissions of the decay chains of U-235, U-238, and Th-232 is presented in this paper because extensive lists have been published in elsewhere [37,38].

### 4.2. Quality Assurance

To perform quality assurance of the calibration curves obtained in this work, three sources (different from the calibration sources) were used. Using Table 5, mean energies were calculated for each peak and compared with the certificate peak energies. Maximum and minimum relative differences between mean peak energies were 2.8% and 1.3% and the average relative difference was 1.7%. Table 5 presents the results for the quality assurance of the verification of the calibration in efficiency. The relative differences between the certificate activity and the activity calculated from measurements with the quality assurance radiation sources were, on average, 5.3%, with minimum and maximum values of 0.5% and 9.9%, respectively.

The calibration quality assurance for the CZT detector for the geometry in question was successfully concluded. Quality assurance of a calibration is an essential step not only to verify if the calibration was performed correctly but also to study the accuracy of the application of the calibration curves. Calculations and experiments analogous to the ones performed in this work can be performed in a field scenario to identify the sources present, as well as their activity.

### 4.3. CZT Sensitivity Analysis

Figure 7 shows a graph of the efficiency for the acquisition at 10 cm and 100 cm SDD. In general, efficiency values tend to be lower when the SDD is higher. For 100 cm SDD, only three peaks could be identified in the graph, namely 59, 121, and 244 keV. At SDD equal to 200 cm, there was no visible difference between the foreground and the background spectra.

Figure 8 shows a graph of efficiency as a function of energy and acquisition time, for different acquisition times. In general, using shorter acquisition times, fewer peaks can be observed in the acquired spectra. For acquisition times of 1 and 2 min, only peaks with mean energy lower than 400 keV can be observed, while, for acquisition time of 5 min, the Cs-137 peak (662 keV) can already be observed. Only for acquisition times of 10, 20 and 30 min can peaks over 662 keV be observed. Moreover, the acquisition time also influences the total number of detected peaks. At lower acquisition times (<5 min), only three peaks are identifiable in the acquired spectra. However, at higher acquisition times, six, five, and seven peaks can be observed for acquisition times of 10, 20, and 30 min, respectively. In general, using shorter acquisition times, two observations can be drawn: (1) fewer peaks can be observed in the acquired spectra; and (2) only peaks with lower average energy can be observed.

When comparing Figure 6 and Figure 7, it is noticeable that the shorter acquisition time in Figure 7 (by a factor of approximately 48) entails higher uncertainty values for the efficiency. In addition, when considering shorter acquisition times, there are less identifiable peaks in the acquired spectra. Regarding Figure 8, two observations can be drawn when shorter acquisition times are used: (1) fewer peaks can be observed in the acquired spectra; and (2) only peaks at lower average energies can be observed. The observed peaks at acquisition times of 10 to 30 min are approximately half of the identifiable peaks in the CZT calibration spectra. However, the average energies of the peaks acquired at 30 min range between 59 and 1173 keV, which may be enough to estimate the activity of the radiation sources present in a field scenario. Moreover, in Figure 8, the uncertainty values rise when the acquisition time is shorter, because the CZT detects fewer counts, which leads to higher uncertainty for each channel/energy. Although the peak at 1332 keV is only seen at the 10 min acquisition time, it should also have been identifiable in the 20 and 30 min spectra. This inaccuracy is attributed to the uncertainty in the number of counts of the peak acquired when performing the measurements.

### 4.4. GMC Verification

Table 6 and Table 7 report the results of the measurements with the GMCs at three different distances from three radiation sources. The tabulated equivalent dose rate values for each source at each distance are only inserted in Table 6 because they are the same for both GMCs. According to Table 6 and Table 7, the average CPM of the set of measurements for both GMCs increases. In addition, the standard deviation and coefficient of variation decrease at shorter distances and when using sources with higher activity. Moreover, while at higher count rates (i.e., lower SDD and using a source with higher activity), the coefficients of variance are similar for both GMCs, at lower count rates (i.e., for the 74 MBq source) the coefficients of variance are higher—by a factor of approximately 2—for the Sparkfun.

Figure 9 shows plots of the equivalent dose rate value (μSv/h) as a function of the count rate of each GMC for all source-distance configurations. Microsoft Excel${}^{TM}$ software was used to obtain a least-squares linear fit of the data. The interception with the yy-axis was set to zero, in order to avoid negative dose rate values for very low count rate values. The linear regressions show that the conversion coefficient varies linearly with the count rate and the equation of the fit can be used to calculate the dose rate as a function of the acquired count rate (Equations (Equation 5) and (Equation 6)).
(5)$$\begin{array}{c} \mathrm{Sparkfun}:\;\mathrm{Dose}\;\mathrm{Rate}\left(\mu Sv/h\right)=0.0134\cdot CPM\left(R^{2}=0.9981\right) \end{array}$$
(6)$$\begin{array}{c} \mathrm{Mazur}:\;\mathrm{Dose}\;\mathrm{Rate}\left(\mu Sv/h\right)=0.0024\cdot CPM\left(R^{2}=0.9985\right) \end{array}$$

The validation procedure was undertaken by placing both GMCs at three different distances from three sources with different activities. As expected, the average count rate for both GMCs increases and the standard deviation and coefficient of variation decrease at shorter SDD, as well as when using sources with higher activity. Considering higher count rates (i.e., sources of 740 and 7400 MBq), the coefficients of variation are much lower and considered to be sufficiently low for the application considered in this work (i.e., radiological mapping). For example, coefficients of variation for the 7400 MBq source are, on average, much lower than for the 74 MBq source, by a factor of approximately 10 and 20 for the Mazur and the Sparkfun, respectively. This effect was due to the higher flux of particles the sensors are able to detect at shorter SDD and with higher activity sources, which raises the count rate and lowers the standard deviation and uncertainty of the measurements.

However, at lower count rates (i.e., higher SDD and using lower activity sources), the coefficients of variation are, in general, lower for the Mazur than for the Sparkfun (by a factor of approximately 2). While for the Mazur the coefficients of variation are acceptable and frequent for this type of detectors, the Sparkfun reports a coefficient of variation with a high value and outside of the expected range of values (i.e., 53%). Additionally, the count rate is consistently higher for the Mazur when compared to the Sparkfun, by an average factor of 4.5. The Mazur is clearly much more sensitive than the Sparkfun, being able to detect more events per unit time and consequently lowering the uncertainty of the measurements. This probably happens because the Mazur is a much more recent and state-of-the-art equipment recommended for NORM detection, while the Sparkfun is an older equipment which is no longer for sale and has been replaced by other equipment. Furthermore, the tube size and sensitive volume are much larger for the Mazur than for the Sparkfun.

The conversion coefficient (in μsV/h/CPM) was calculated for both GMCs considering three different sources and three different distances to source. According to Figure 9, the linear regressions represent a very good fit to the data (very high R2 value) and the conversion factor varies linearly with the count rate. Therefore, Equations (Equation 5) and (Equation 6) can be used to carry out the conversion from count rate to dose rate for the Sparkfun and the Mazur, respectively.

Moreover, the Mazur manual [36] reports its sensitivity to be 3500 CPM/mR/h ≈ 350 CPM/μSv/h. If this number were inverted, the correction factor would be $2.85\times 10^{3}$μSv/h/CPM, which is within the range of the conversion factors obtained in this study considering various radiation sources and SDDs.

### 4.5. GMC Sensitivity Analysis

The dose rate for each GMC was calculated using Equations (Equation 5) and (Equation 6) from the linear regressions presented in Figure 9. Figure 10 shows plots of the dose rate of each GMC as a function of the SDD and the acquisition time. Considering both GMCs, the dose rate varies much more with the SDD than with the acquisition time, which is supported by three observations in Figure 10: (1) the dose rate decreases exponentially with the distance to the radiation source, according to the inverse-square law of the distance; (2) as the acquisition time increases, the dose rate values tend to converge to a real value; and (3) in general, the variation of the dose rate values for the same SDD is lower than the variation of the dose rate values between SDDs. In general, using acquisition times over 5 or 10 min, the measured values start to stabilize and do not vary significantly when compared to the highest acquisition time (i.e., 60 min). Moreover, there is a higher variance along the acquisition time in the dose rate values measured with the Sparkfun, when compared with the Mazur, and in the values measured at higher distances.

Regarding the response time measurements, according to the criteria described in Section 3.5, each sensor was considered to have “detected” the sources when the count rate was higher than 1202 CPM and 205 CPM for the Sparkfun and the Mazur, respectively. Figure 11 shows the response time for each measurement considering both GMCs. For the Sparkfun, the average response time was 5.08 s, with a standard deviation of 1.62 s and a coefficient of variation of 31.9%. For the Mazur, the average response time was 4.36 s, with a standard deviation of 1.30 s and a coefficient of variation of 29.8%. Considering the shorter time interval of 1 s, the Sparkfun was considered to have “detected” the sources almost always in the next measurement after the sources were placed in front of it. In this setup, the response time was, on average, approximately 1 s, with a maximum value of 2 s.

The results of the sensitivity analysis of the GMCs show that the acquired quantity is much more dependent on the SDD than on the acquisition time, which was expected, namely because the dose rate decreases exponentially with the SDD and the variance in dose rate for different acquisition times is, in general, not too high. Moreover, using acquisition times over 5–10 min, the measured values are considered, in general, accurate enough for the purpose of this work. Furthermore, the higher variance for the dose rate along the acquisition time with the Sparkfun (compared with the Mazur) and in the values measured at higher SDDs is explained above in Section 4.4. The average response time is close for both GMCs, but slightly higher for the Sparkfun, namely because of the lower sensitivity when compared to the Mazur, which is due to the reasons already explained in Section 4.4. However, when considering a time between measurements of 1 s for the Sparkfun, the response time decreases by factor of approximately 5, which increases the GMC sensitivity.

### 4.6. Recommendations and Flight Strategy for Radiological Monitoring Using UAVs

As mentioned in the Introduction, the global flight strategy for radiological monitoring and radiation mapping studied in this work comprised two stages: (1) generic monitoring, where GMCs on board of UAVs are used to scan the area of interest, and identify radiation *hotspots*; and (2) *hotspot* inspection, where a UAV with a CZT lands on each *hotspot* to verify the presence and identify and estimate the activity of possible radioactive sources.

Generic monitoring should be performed in a field scenario at 1–2 m from the ground, as a compromise between the height of the vegetation and the GMC sensitivity. The verification of the GMCs was performed in this range of SDDs. Linear equations were reported for conversion from the count rate (in CPM) provided by the GMCs to equivalent dose rate. To perform dose rate mapping, Equations (Equation 5) and (Equation 6) can be used to carry out this conversion for the Sparkfun and the Mazur, respectively. The results of the sensitivity analysis of the GMCs showed that the acquired quantity is much more dependent on the SDD than on the acquisition time. Moreover, for acquisition times over 5–10 min, the measured values are considered, in general, accurate. This finding will be very useful, because the confirmation of the existence of a *hotspot* found during generic monitoring can be performed by a UAV with a GMC in a relatively short hover time (i.e., 5–10 min).

In addition, the response time measurements showed that the minimum hover time for the GMCs to detect a *hotspot* is 5.08 and 4.34 s for the Sparkfun and the Mazur, respectively. Considering a square with 1 m side, in order for a UAV to detect a *hotspot*, it would have to hover the square during at least the response time, and thus the minimum speed for the UAV would be 0.20 and 0.23 m/s for the Sparkfun and the Mazur, respectively. These are realistic UAV speeds for use in a field scenario. Moreover, the lower speeds contribute to a more accurate control on the UAV trajectory. However, if the detector response time decreased by a factor of 5 (i.e., to approximately 1 s), then the minimum UAV speed would increase to 1 m/s, thus decreasing the time necessary to perform radiological mapping of a given area.

In similar studies, radiological mapping was performed with UAVs flying at speeds mainly between 1 and 1.5 m/s, and between 1 and 10 m from the ground [13,14,15,17].

However, these studies performed radiological mapping in areas where radiological accidents happened, such as near Fukushima, where higher activity sources exist. Martin et al. [4] used a UAV coupled with a CZT to perform radiological monitoring in out-of-commission uranium mines, flying at 1.5 m/s and at 5–15 m from the ground, due to the vegetation existing in the area (i.e., the treeline).

The speed recommended in this work is better suited for flying at lower altitudes, because possible obstacles are more easily avoided and flying at lower altitude allows for mapping with higher spatial resolution and more accurate radiological measurements.

Regarding the second flight strategy, since the gamma radiation emitted by the decayed species of U-235, U-238, and Th-232 are predominantly between 59 and 1173 keV, according to the conditions presented in this study, the spectra acquisition with the CZT should be performed at a short SDD, between 10 and 100 cm.

Given that the UAV with the CZT will land on the ground where a *hotspot* is detected, a source–detector distance between 10 and 20 cm is feasible. Preferably, the acquisition should be performed as close as possible, and in no less than 30 min, in order to acquire spectra with the maximum number of peaks and with a higher number of counts, This will be reflected into lower uncertainties. Despite several other studies having used UAVs coupled with CZTs for radiological monitoring [4,6,14,15,17], the CZTs were used to scan the area, instead of functioning as a precise measurement tool to assess the ionizing radiation spectra in small *hotspot* areas identified by other detectors.

The protocol to apply after performing field measurements with the CZT is the following: (1) use Equation (Equation 1) to convert channel number to energy and the mean energy of the peaks present in the spectrum to identify possible radionuclides present in the *hotspot*; (2) calculate the ratio of the peak area of the acquired spectrum and the CZT efficiency at the peak mean energy (efficiency can be calculated using Equations (Equation 3) and (Equation 4)); and (3) determine the activity of the source present in the *hotspot* by dividing the result by the respective yield on the decay scheme of the radionuclide.

It should be noted that the results reported in this work regarding the use of sensors for radiological monitoring were based on experiments at idealized distances and geometries and using the available sources. An effort was made to perform experiments in similar conditions to the ones that would exist in a field measurement (e.g., choosing the most adequate sources and SDDs) and thus the findings and recommendations are considered valid for the reported conditions.

## 5. Conclusions and Future Work

For the purpose of undertaking extensive radiological monitoring of out-of-commission uranium mine territories, the output of this work provides valuable information to the flight strategy, namely about the physical parameters of the drone flight and physical quantities conversion for data processing and interpretation. In each stage of the flight strategy, the agents of the fleet are equipped with different sensors, have different missions, and acquire different data, cooperating and communicating between them to map the area of interest. Table 8 summarizes the flight strategy. The agents in the proposed flight strategy (Table 8) cooperate by providing information used to plan the flight of the others and/or contribute to the construction of the radiation map of the area of interest. Furthermore, it is possible for all the drones to fly at the same time while exchanging information; for example, as the GMC Drone is scanning a scenario and finds radiation hotspots, the CZT Drone can be simultaneously landing in the hotspots and acquiring the spectra.

Figure 12 shows a CAD model of the UAV developed and assembled in the scope of the FRIENDS project coupled with the Sparkfun and CZT sensors. Figure 13 shows the UAV flying. In the near future, the flight strategy and recommendations proposed in this work will enable progressing to the next step of the project FRIENDS, consisting on the radiological monitoring of continental areas where radioactivity levels are expected to overcome the natural background, due to NORM industries or radiological events. As a proof of concept, the prepared flight strategy will be used to survey out-of-commission uranium mines in northern Portugal.

Considering the proximity of these mines to communities and agricultural villages, the accurate identification and quantification of the natural radionuclides is of major importance in order to prevent any further risks of exposure to hazardous ionizing radiation to the population and the environment.

## Figures and Tables

**Figure 1 sensors-20-01538-f001:**
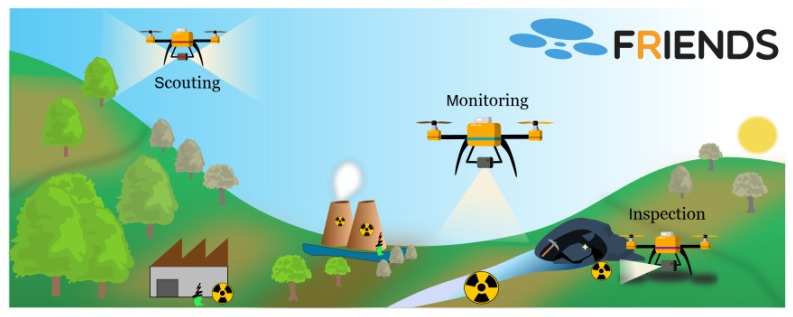
The FRIENDS mission is focused on three challenges.

**Figure 2 sensors-20-01538-f002:**
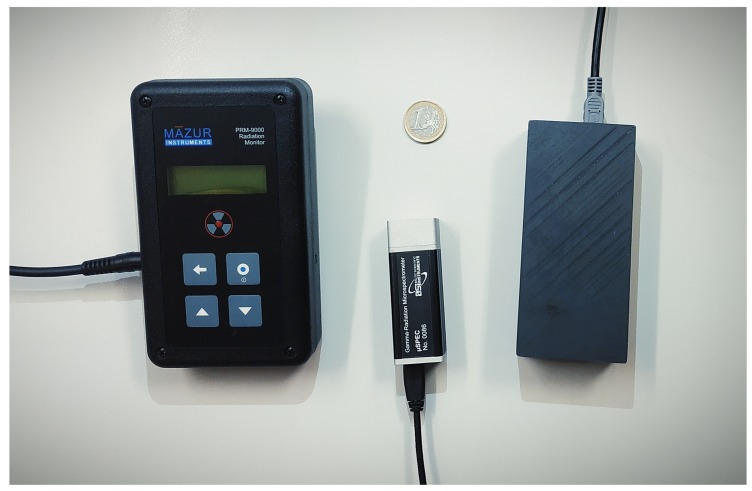
The radiological sensors used in this work. From left to right: the Mazur PRM-9000, the μSPEC 500 CZT, and the Sparkfun SEN-11345. A 1 euro coin is also present for scale.

**Figure 3 sensors-20-01538-f003:**
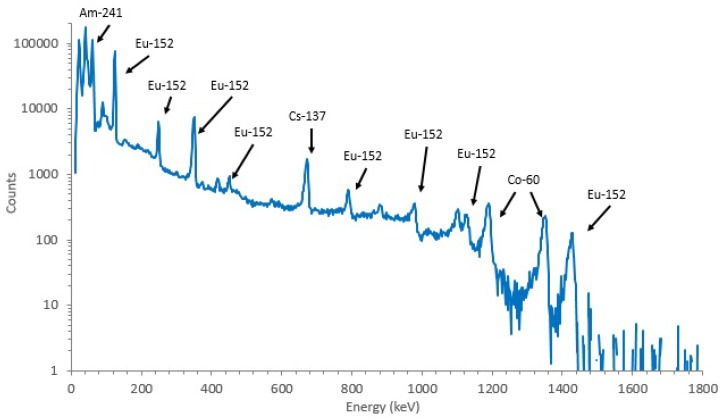
Acquired calibration spectrum (background subtracted). Labeled peaks were considered for calibration. Time of acquisition was 86,400 s.

**Figure 4 sensors-20-01538-f004:**
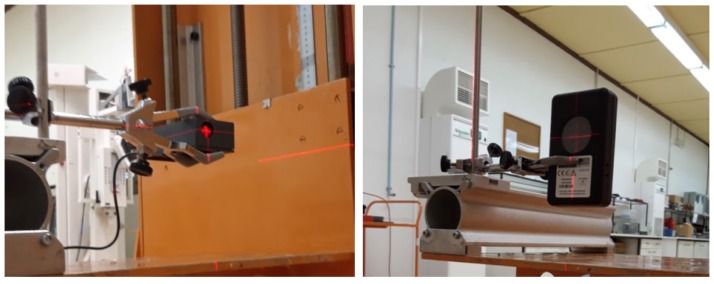
Acquisition setup for the Sparkfun (**left**) and the Mazur (**right**) verification procedure. The geometric center of the sensitive volume of each GMC was aligned with the source geometric center using vertical and horizontal lasers.

**Figure 5 sensors-20-01538-f005:**
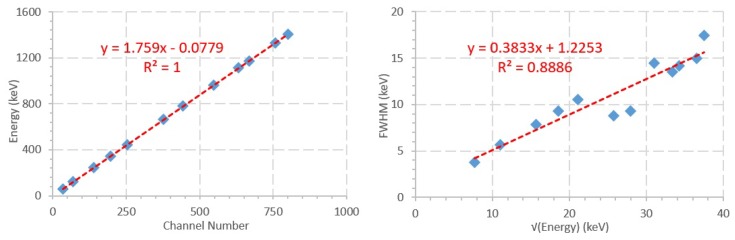
Energy (**left**) and FWHM (**right**) calibration of the considered peaks, showing the linear regression equations and R value.

**Figure 6 sensors-20-01538-f006:**
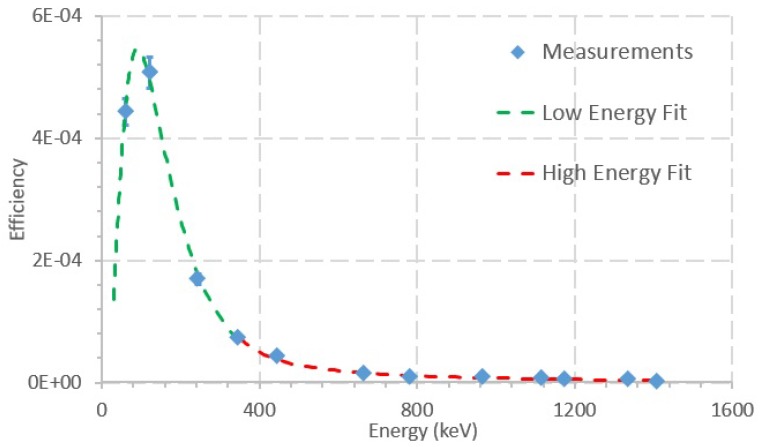
Peak efficiency as a function of the energy for every considered peak. The graph depicts the measurements for each peak in blue and the Low Energy Fit (second-degree polynomial fit, up to 344.3 keV, green) and the High Energy Fit (third-degree polynomial fit, above 344.3 keV, red).

**Figure 7 sensors-20-01538-f007:**
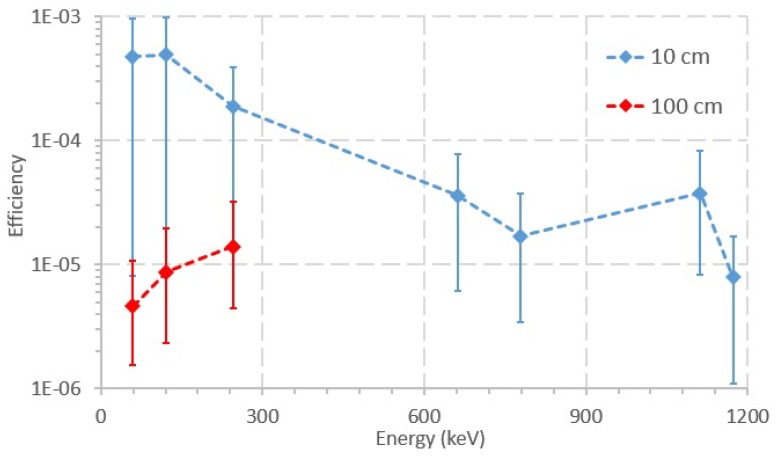
Efficiency of the photopeaks acquired at 10 and 100 cm. The acquisition time was 1800 s.

**Figure 8 sensors-20-01538-f008:**
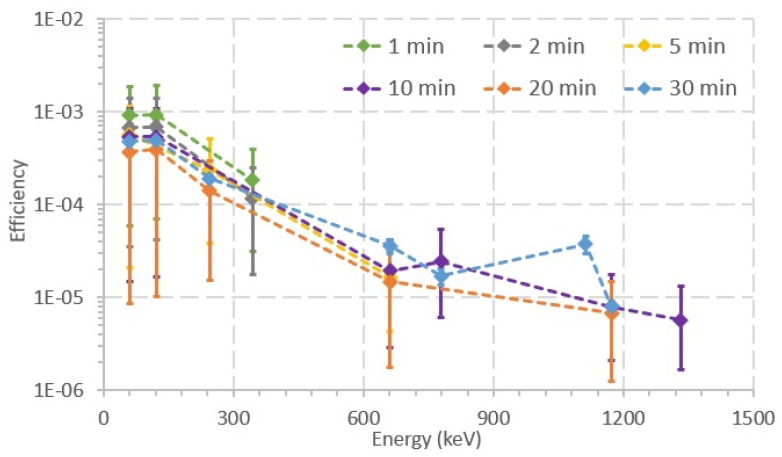
Efficiency of the photopeaks observed at different acquisition times, with the same geometry used in the calibration. Lines and markers of different colors represent peak efficiencies with different acquisition times, ranging between 1 and 30 min.

**Figure 9 sensors-20-01538-f009:**
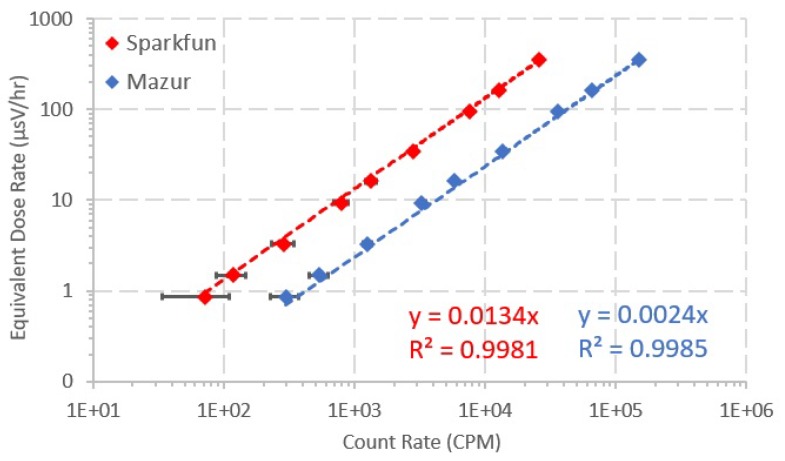
Equivalent dose rate as a function of the count rate of the two GMCs. Nine data points were retrieved for each GMC, representing each data acquisition, one for each source-distance configuration. The least-squares linear fit of the data was calculated and is also presented.

**Figure 10 sensors-20-01538-f010:**
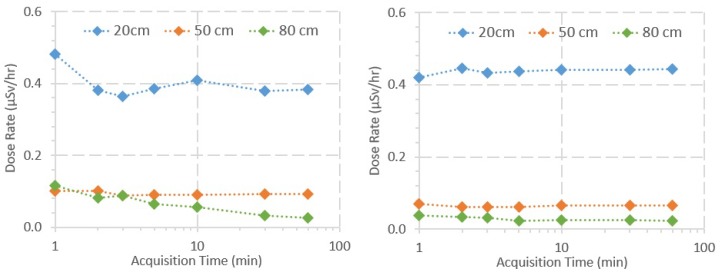
Dose rate as a function of the acquisition time for measurements using the Sparkfun (**left**) and the Mazur (**right**) at 20, 50, and 80 cm from two Cs-137 sources.

**Figure 11 sensors-20-01538-f011:**
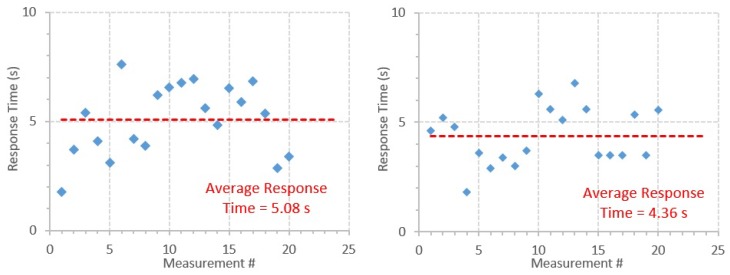
Detector response time for each measurement exposing the Sparkfun (**left**) and the Mazur (**right**) detectors to two Cs-137 sources.

**Figure 12 sensors-20-01538-f012:**
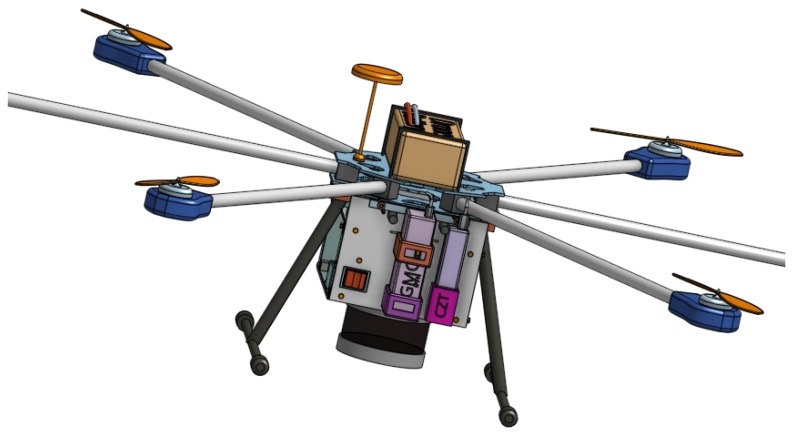
CAD image of the hexarotor and sensor-box underneath, encompassing GMC and CZT sensors as well as a LIDAR on the bottom.

**Figure 13 sensors-20-01538-f013:**
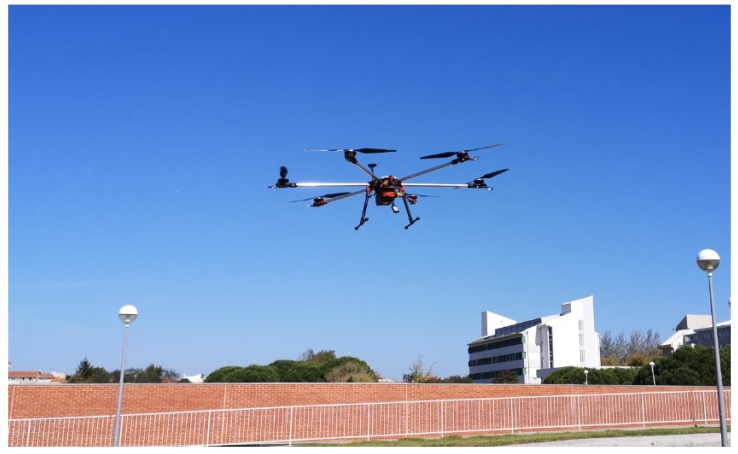
Preliminary tests of the UAV flying over a park.

**Table 1 sensors-20-01538-t001:** CZT characteristics as declared by the manufacturer.

Parameter	Value
Sensitive volume	Crystal CdZnTe, 500 mm${}^{3}$
Energy Range	20.0 keV to 3.0 MeV
Energy Resolution	<2.5% FWHM @ 662 keV
Dimensions	25 × 25 × 72 mm${}^{3}$
Weight	80 g

**Table 2 sensors-20-01538-t002:** GMC characteristics as declared by the manufacturers [35,36].

Parameter	Sparkfun	Mazur
Power Supply	USB, 30 mA, 5 V	USB, 100 mA, 9 V
Sensitive Volume/Tube	LND 712	LND 7317 pancake
Dimensions	105 × 44 × 25 mm${}^{3}$	143 × 83 × 35 mm${}^{3}$
Weight	50 g	378 g
Measurement Range	unknown	1 to 437500 CPM

**Table 3 sensors-20-01538-t003:** Characteristics of the peaks emitted by the Eu-152 and radionuclide mix calibration sources. In total, 12 peaks, marked with a “*” next to the peak energy, were considered for calibration.

Nuclide	Energy (keV)	$t_{\frac{1}{2}}$ (d)	Yield (%)	Initial Activity (γ/s)
Am-241	59.5 *	157,860.0	0.357	6.85$\;\times 10^{3}$
Cd-109	88.0	462.6	0.036	9.34$\;\times 10^{3}$
Eu-152	121.8 *	4943.0	0.286	3.05$\;\times 10^{4}$
Co-57	122.1	271.8	0.856	5.11$\;\times 10^{3}$
Ce-139	165.9	137.6	0.800	7.21$\;\times 10^{3}$
Eu-152	244.7 *	4943.0	0.076	3.05$\;\times 10^{4}$
Hg-203	279.2	46.61	0.810	1.53$\;\times 10^{4}$
Eu-152	344.3 *	4943.0	0.266	3.05$\;\times 10^{4}$
Sn-113	391.7	115.1	0.640	1.00$\;\times 10^{4}$
Eu-152	443.9 *	4943.0	0.310	3.05$\;\times 10^{4}$
Cs-157	661.7 *	10,983.0	0.851	6.60$\;\times 10^{3}$
Eu-152	778.9 *	4943.0	0.129	3.05$\;\times 10^{4}$
Y-88	898.0	106.6	0.937	2.41$\;\times 10^{4}$
Eu-152	964.1 *	4943.0	0.146	3.05$\;\times 10^{4}$
Eu-152	1112.1 *	4943.0	0.136	3.05$\;\times 10^{4}$
Co-60	1173.2 *	1925.4	0.997	1.21$\;\times 10^{4}$
Co-60	1332.5 *	1925.4	0.999	1.21$\;\times 10^{4}$
Eu-152	1408.0 *	4943.0	0.210	3.05$\;\times 10^{4}$
Y-88	1836.8	106.6	0.992	2.55$\;\times 10^{4}$

**Table 4 sensors-20-01538-t004:** Characteristics of the peaks considered for calibration retrieved from the spectrum acquired with the CZT.

Nuclide	Ch No.	Mean Peak Energy (keV)	FWHM (keV)	Efficiency ϵ	Uncertainty of ϵ (%)
**Am-241**	33.85	59.47	3.79	4.43$\;\times 10^{-4}$	0.24%
**Eu-152**	69.25	121.73	5.68	5.07$\;\times 10^{-4}$	0.26%
**Eu-152**	139.1	244.60	7.82	1.69$\;\times 10^{-4}$	1.16%
**Eu-152**	195.81	344.35	9.30	7.30$\;\times 10^{-5}$	0.80%
**Eu-152**	252.52	444.11	10.55	4.21$\;\times 10^{-5}$	0.61%
**Cs-157**	376.65	662.45	8.81	1.48$\;\times 10^{-5}$	1.51%
**Eu-152**	443.35	779.79	9.28	1.06$\;\times 10^{-5}$	3.88%
**Eu-152**	547.81	963.53	14.48	8.61$\;\times 10^{-6}$	4.24%
**Eu-152**	632.41	1112.35	13.52	6.36$\;\times 10^{-6}$	5.08%
**Co-60**	667.23	1173.58	14.20	5.35$\;\times 10^{-6}$	2.55%
**Co-60**	757.22	1331.88	14.95	4.07$\;\times 10^{-6}$	2.62%
**Eu-152**	800.02	1407.17	17.43	3.76$\;\times 10^{-6}$	3.50%

**Table 5 sensors-20-01538-t005:** Comparison of the certificate activity (corrected for the time of measurement) and the activity calculated from measurement with the verification sources.

Source	Mean Peak Energy (keV)	Corrected Certificate Activity (Bq) ± Uncertainty (%)	Measured Activity (Bq) ± Uncertainty (%)	Relative Difference (%)
**Pb-210**	46.54	4062.2 ± 1.6%	4212.1 ± 8.6%	3.7%
**Cs-137**	661.66	14,474.3 ± 1.0%	14,619.7 ± 3.6%	0.5%
**Co-60**	1173.23	12,636.6 ± 0.7%	13,396.0 ± 3.8%	7.0%
**Co-60**	1332.49	12,636.6 ± 0.7%	13,414.6 ± 4.0%	9.9%

**Table 6 sensors-20-01538-t006:** Average value, standard deviation, coefficient of variation, and conversion coefficient of each acquired set of 20 values for the Sparkfun, considering each source-distance configuration. Background acquisition had an average value of 16.8 CPM and a standard deviation of 12.6 CPM.

Source Activity (mCi)		2			20			200	
**Distance to Source (m)**	2	1.5	1	2	1.5	1	2	1.5	1
**Equivalent Dose Rate (μSv/hr)**	0.86	1.50	3.28	9.33	16.14	34.94	93.56	162.09	351.67
**Average (CPM)**	71.4	117.6	286.2	791.4	1336.4	2793.0	7646.4	12,717.6	25,743.6
**Standard Deviation (CPM)**	37.8	30.1	53.6	94.9	115.2	147.5	167.9	232.6	457.1
**Coefficient of Variation (%)**	53%	26%	19%	12%	9%	5%	2%	2%	2%
**Conversion Coefficient (μSv/hr/CPM)**	1.21$\;\times 10^{-2}$	1.28$\;\times 10^{-2}$	1.15$\;\times 10^{-2}$	1.18$\;\times 10^{-2}$	1.21$\;\times 10^{-2}$	1.25$\;\times 10^{-2}$	1.22$\;\times 10^{-2}$	1.27$\;\times 10^{-2}$	1.37$\;\times 10^{-2}$

**Table 7 sensors-20-01538-t007:** Average value, standard deviation, coefficient of variation, and conversion coefficient of each acquired set of 20 values for the Mazur, considering each source-distance configuration. Background acquisition had an average value of 44.8 CPM and a standard deviation of 23.8 CPM.

Source Activity (mCi)		2			20			200	
**Distance to Source (m)**	2	1.5	1	2	1.5	1	2	1.5	1
**Average (CPM)**	298.8	537.3	1257.1	3278.7	5860.9	13,670.4	35,950.9	65,683.8	150,970.1
**Standard Deviation (CPM)**	70.6	85.1	85.9	202.9	226.0	558.6	1244.9	1471.4	2199.9
**Coefficient of Variation (%)**	24%	16%	7%	6%	4%	4%	3%	2%	1%
**Conversion Coefficient (μSv/hr/CPM)**	2.89$\;\times 10^{-3}$	2.80$\;\times 10^{-3}$	2.61$\;\times 10^{-3}$	2.85$\;\times 10^{-3}$	2.75$\;\times 10^{-3}$	2.56$\;\times 10^{-3}$	2.60$\;\times 10^{-3}$	2.47$\;\times 10^{-3}$	2.33$\;\times 10^{-3}$

**Table 8 sensors-20-01538-t008:** Mission summary and physical parameters for each agent present in each Stage of the proposed flight strategy.

Stage	Agent	Mission	Distance to Ground	Mission Time per 0.5 ha (Football Field)	Flight Speed
**0**	Base Station	Define Area of Interest & dispatch drones	-	-	-
**1**	LIDAR Drone	Perform 3D reconstruction of the scenario	10–15 m	7 min	2 m/s
**2**	GMC Drone	Sweep the scenario to detect hotspots	1–2 m	45 min	Mazur: 0.2 m/s, Sparkfun: 0.23 m/s
**3**	GMC Drone	Hop across identified hotspots to acquire accurate equivalent dose rates	10–20 cm	Mazur, Sparkfun: 5–10 min (per hotspot)	2 m/s
**4**	CZT Drone	Hop across identified hotspots to acquire the spectra of the sources	10–20 cm	30 min (per hotspot)	2 m/s

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
