# Peer review of "Performance Analysis of Geiger–Müller and Cadmium Zinc Telluride Sensors Envisaging Airborne Radiological Monitoring in NORM Sites"

_sensors, 2020, doi:10.3390/s20051538_

Round 1
Reviewer 1 Report
The work is focused on the quality assurance, sensitivity analysis, and calibration of radiological detectors using GMC and CZT sensors to be used with autonomous UAVs to perform radiological monitoring and mapping in areas with high concentrations of naturally occurring radioactive materials (NORM). The work presents experimental results of energy and efficiency calibration for both types of sensors. The results are very important to parametrize the use of drones for general radiological monitoring and hot spots localization and quantification. The main focus of the work is on sensors characterization and identification. The authors should consider giving more details for the conclusions regarding the drone flight mission and physical parameters such as distance to the ground, acquisition time and operation speed for both types of sensors (GMC and CZT), and especially the question regarding the cooperation aspects for monitoring with a fleet of drones.
Minor corrections are needed such as in line 62, pg 2 (word onde), and line 268, pg 9 (repeated word the).
Author Response
Please see the attached document. The reviewer comments are presented in black colored letters and the responses of the authors in purple colored letters.

Reviewer 2 Report
Reviewer's comments on " Application of Geiger-Müller and Cadmium Zinc Telluride sensors for radiological monitoring using drones" by Jorge Borbinha et al.
This paper studies the use of radiation detectors such as Geiger-Muller Counters (GMC) and Cadmium Zinc Telluride (CZT) spectrometers in preparation for future field measurements. It is claimed that the proposed methods and results can be used to formulate the requirements for a UAV flight strategy, which will be used to survey out-of-commission uranium mines in northern Portugal. Several tests were conducted to assure the calibration of the CZT and verification of the GMCs. Also, a sensitivity analysis of the sensors was carried out considering different acquisition times, distance to source and detector response time. The manuscript reported specific requirements, such as UAV distance to the ground, time of exposition, speed and the methodology to perform the identification and calculate the activity of possible radioactive sources. The authors insisted that an effective flight strategy is also presented, aiming to use radiation detectors coupled with UAVs to undertake extensive monitoring of areas with enhanced levels of environmental radiation, which is of prime importance due to the lasting hazardous effects of enhanced environmental radiation in the nearby ecosystem and population.
First of all, the title of the current manuscript should be revised as no methods or results are found for the use of drones to monitor field radiation. Most parts of the manuscript focus only on the calibration, sensitivity analysis, and verification of CZT and GMC sensors. The conceptual drawing of a drone combined with these sensors is presented on the last page with the actual drone figure. Although it is described that an effective flight strategy is also presented, aiming to use radiation detectors coupled with UAVs to undertake extensive monitoring of areas with enhanced levels of environmental radiation, no flight strategy is presented. The current title may misguide readers that the contribution of the paper is to provide methods and results to monitor field radiation by using drones, which is not the case. Thus, it is strongly recommended to change the title, to best represent the contribution of the manuscript.
Secondly, and more importantly, it is not clear what the major contribution of the paper is. As the reviewer already pointed out, the manuscript provided the calibration, sensitivity analysis, and verification of two sensors. However, it is also described by authors on page 16 that “The efficiency curve in this work is very similar in shape to the other published efficiency curves…” To distinguish the contribution of the manuscript from that by other published papers, the contents should include the radiological monitoring results by combining two sensors (e.g., what method can be used to combine two sensors and how it improves the result). The current manuscript only provided some results for individual sensor analysis.
A minor comment is that it would be better to include “Discussion” in the “Results” section. This is mainly because “Results” contains too many different results so that the reader may not follow the analysis and discussion if separated.
Author Response
Please see the attached document. The reviewer comments are presented in black letters and the responses of the authors in purple colored letters.

Round 2
Reviewer 2 Report
The authors answered all comments and questions raised by the reviewer in the revised manuscript and the answer letter.